# Integration of a clinical pharmacist workforce into newly forming primary care networks: a qualitatively driven, complex systems analysis

Thomas Mills ,[1,2] Mary Madden ,[1] Duncan Stewart,[3] Brendan Gough,[4] Jim McCambridge[1]

¹Department of Health Sciences, University of York, York, UK
²PHIRST South Bank, London South Bank University, London, UK
³Centre for Primary Health and Social Care, London Metropolitan University, London, UK
⁴Leeds Beckett University, Leeds, UK

**Correspondence to**
Dr Thomas Mills;
millst3@lsbu.ac.uk

## ABSTRACT

**Objective** The introduction of a new clinical pharmacist workforce via Primary Care Networks (PCNs) is a recent national policy development in the National Health Service in England. This study elicits the perspectives of people with responsibility for local implementation of this national policy package. Attention to local delivery is necessary to understand the contextual factors shaping the integration of the new clinical pharmacy workforce, and thus can be expected to influence future role development.

**Design** A qualitative, interview study

**Setting and participants** PCN Clinical Directors and senior pharmacists across 17 PCNs in England (n=28)

**Analysis** Interviews were transcribed, coded and organised using the framework method. Thematic analysis and complex systems modelling were then undertaken iteratively to develop the themes.

**Results** Findings were organised into two overarching themes: (1) local organisational innovations of a national policy under conditions of uncertainty; and (2) local multiprofessional decision-making on clinical pharmacy workforce integration and initial task assignment. Although a phased implementation of the PCN package was planned, the findings suggest that processes of PCN formation and clinical pharmacist workforce integration were closely intertwined, with underpinning decisions taking place under conditions of considerable uncertainty and workforce pressures.

**Conclusions** National policy decisions that required General Practitioners to form PCNs at the same time as they integrated a new workforce risked undermining the potential of both PCNs and the new workforce. PCNs require time and support to fully form and integrate clinical pharmacists if successful role development is to occur. Efforts to incentivise delivery of PCN pharmacy services in future must be responsive to local capacity.

## INTRODUCTION

Reform efforts globally have sought to develop the primary care sector with the underpinning idea that the sector is vital to contemporary healthcare challenges.[1] In the UK, a range of institutional reforms have, over the past four decades, been enacted to transform the traditional set-up of small private business

### STRENGTHS AND LIMITATIONS OF THIS STUDY

⇒ The present study represents a rigorous and timely qualitative investigation of the early integration of a new clinical pharmacist workforce into primary care that is anticipated to take on a significant, patient-facing role in future.

⇒ The sampling approach allowed the research to capture insights on the new clinical pharmacy workforce across 17 diverse Primary Care Networks (PCNs) in England.

⇒ While the diversity of PCNs provided broad brush insight, a case study approach that focused on a smaller number of PCNs more in depth could provide deeper lessons.

⇒ The principal study focus on the clinical pharmacist workforce means that the study does not explore other developments that are being implemented as part of the PCN policy package.

⇒ The study focus on senior staff (PCN Clinical Directors and senior pharmacists) could also be complemented, in future, by the qualitative study of the perspectives of the newly integrated workforce itself. Broadening the sample to include other general practice staff, for example, practice managers and nursing staff, would also allow exploration of diverse perspectives and multidisciplinary working.

partnerships run by General Practitioners (GPs). GPs have traditionally been contracted by the National Health Service (NHS) to provide a range of services to a local population and coordinate patient care, including referrals to hospital-based specialist care. Reforms to this set-up have included attempts to enhance the role of GPs in healthcare commissioning and new models of service delivery that have transferred services out of hospitals into primary care.[2–4] There have also been efforts to address shortfalls in the numbers of GPs and meet new healthcare challenges, including facing the increasing demand and complexity of patient needs, through the integration and development

of new primary care roles.[5–7] This reflects a broader task shifting to 'mid-level' professionals who are educated and trained to a level where they can undertake some of the activities of GPs and specialists within a prescribed scope of practice.[7 8] In primary care, some GP work tasks and responsibilities have shifted to nursing staff[5] and, more recently, clinical pharmacists.[6 9] Such developments occur within ongoing contexts of reorganisations, funding constraints and other pressures.[10]

The NHS Long Term Plan, published in January 2019, featured a new funding settlement for primary care (with projected funding to be £4.5 billion higher in 2023/2024 than in 2019/2020) and plans for new staff roles and services.[11 12] In order to qualify for new funding, GPs were to collaborate with neighbouring practices to form Primary Care Networks (PCNs).[11 12] The aim was to expand the primary care workforce and enhance patient care through new and improved services, based on aspects of Canadian[13] and New Zealand[14] primary care systems. A new national contract, the Network Contract Directed Enhanced Services (known colloquially as 'the DES'), was introduced on 1 July 2019. This facilitated access to funds for new healthcare roles via the Additional Roles Reimbursement Scheme (ARRS), including social prescribing link workers, physiotherapists, paramedics and pharmacists. A total of 1250 PCNs were formed across England in the first year of the policy, each serving around 30–50 000 patients.[12]

Early research in PCN policy implementation reveals a contrasting picture of initial progress, with governance arrangements established and new staff quickly recruited, amidst significant 'organisational fragility'.[15] A National Institute for Health Research funded study of PCN formation found considerable challenges on set-up due to a short policy time frame and the complexities involved in collaborating across practice boundaries.[16] Qualitative research in this emerging field has explored the consistency of national PCN policy objectives,[17] GP views and early experiences of the PCN policy,[18] local commissioner support[19] and, most recently, the integration of new healthcare roles into PCNs.[20] Given that a central PCN objective is to expand the primary care workforce, there is a clear research need for further research into the integration of PCN healthcare staff and early role development.

This study presents a qualitative investigation of the early integration of a clinical pharmacy workforce into newly forming PCNs. Clinical pharmacists are a key example of a 'mid-level' professional group.[7 8] Historically, pharmacists in the UK have mainly operated in secondary care hospitals or in community pharmacy, providing prescription dispensing services and, more recently, over-the-counter advice.[21] Numbers of pharmacists in GP practices increased during the 1990s via the GP Fundholding scheme and through the 2000s, but the absence of a national workforce strategy resulted in significant diversity, in terms of numbers and roles, across GP practices.[6 21] A key long-term aim of PCN policy was to implement nationally a patient-facing clinical pharmacy role, following an initial pilot in 2016[6]: PCNs were permitted to employ a single clinical pharmacist in their first year of formation (2019), with this number expected to rise to five clinical pharmacists per PCN by 2024.[12] The research aim was to investigate how the PCN policy package was being implemented locally and to develop understanding of the contextual factors that were enabling or constraining the integration of the new clinical pharmacy workforce. A complex systems perspective was adopted that conceives of public policies as multilayered systems characterised by non-linearity, emergence, feedback and adaptations across national, regional and local levels.[22 23] We sought to explore policy emergence and adaptation in diverse PCNs across England using complex systems modelling techniques to model emergent pathways and local-level outcomes.[24 25]

## METHODS

A range of exploratory studies were previously undertaken to assess the potential of the pharmacist role with a view to incorporating attention to alcohol within medicine reviews.[9 26 27] These studies featured within a research programme that started in community pharmacy and then moved into general practice as the new clinical pharmacist workforce was recruited there, via PCNs. This guided the development of this study, including research questions and approach, as we sought to understand and assess the new clinical pharmacy workforce for its suitability to take part in a research trial. The study included interviews with senior staff with responsibility for integrating the clinical pharmacy workforce into newly forming PCNs. The interviews took place between March 2020, a year after PCNs were established, through to September 2021, providing a snapshot of this aspect of the national policy package. Three sets of interviews were undertaken, creating three substudy datasets:

1. Twelve semistructured interviews with seven senior PCN staff (three GP Clinical Directors and four senior PCN pharmacists), between March 2020 and September 2021, in six PCNs based in the Yorkshire and Humber and North-East regions of England (ie, two interviews 1 year apart to explore developments over time, although one participant changed role and declined the second interview). Quotation codes: CD (Clinical Director) and SP (senior pharmacist); the addition of 2 indicates second interview.

2. Ten one-off semistructured interviews between March 2021 and August 2021 with 10 pharmacists in 10 PCNs who were already working in primary care prior to the formation of PCNs and were established in a senior role (either in a GP practice or a PCN) during the transition. Quotation codes: SPX.

3. Six semistructured interviews conducted between September 2020 and June 2021 with three newly appointed SPs across four PCNs (ie, two interviews 1 year apart to explore developments over time; one

interviewee switched to a different PCN during the study). Quotation codes: SPY; the addition of 2 indicates second interview.

The sampling strategy for substudy 1 was pragmatic and opportunistic: we initially sought diversity of PCNs, in terms of operational models and patient populations, but this was not possible to ensure prior to the interviews because PCNs were still forming. Changing PCN characteristics were therefore explored through the interviews. The research team utilised a combination of existing PCN contacts and new contacts, established via telephone calls to GP practices, to recruit CDs and SPs into the substudy. Substudy 2 participants were recruited from PCNs across England to broaden the focus beyond the Yorkshire and Humber and North-East regions. Opportunistic sampling and snowballing techniques were used here. A leaflet describing the study was distributed via national pharmacy organisations and on social media, which included an invite to contact the research team. Substudy 3 included new clinical pharmacists who were recruited via the contacts developed through substudies 1 and 2. Substudy 3 primarily focused on new clinical pharmacists' early training and role but three of the pharmacists (in a sample of 10) were appointed or promoted to a senior pharmacy position during the study and therefore assumed some responsibility for the new clinical pharmacy workforce: data pertaining to PCN formation and clinical pharmacist integration were therefore incorporated into the analysis. All interviewees provided written consent, following an initial contact with the research team, during which the study was discussed and participants were screened for eligibility. In total, 28 senior PCN staff were interviewed across 17 PCNs.

An initial topic guide was created that included questions about PCN formation, clinical pharmacy workforce integration and new, PCN clinical pharmacy services (see online supplemental file). This was piloted with a SP and applied flexibly within the interviews. TM and MM undertook all interviews separately, all except one of which were digitally recorded and transcribed (due to a technical fault but detailed notes were taken immediately after the interview). Interview duration ranged from 42 minutes to 120 minutes. Data analysis proceeded iteratively alongside data collection to allow emerging issues of interest to be identified for further exploration. Data analysis combined the framework method[28] with complex systems modelling.[24 25] Data were organised in NVivo V.12 and summarised in a Microsoft Word document. TM and MM worked together to develop and apply a coding framework, cross checking and engaging in consensus discussion throughout, to ensure the trustworthiness and credibility of the analytic process. The coding framework was applied to all data using NVivo. A narrative summary of each interview was created to ensure the coding process did not strip away the contextual richness of interviewee accounts.[29] Codes that were identified as significant to PCN formation and clinical pharmacist role development were summarised across the dataset, in a form of framework analysis.

An iterative process of analysis, theme development, modelling and group discussion then ensued, involving the research team, senior academics, practitioners and patient representatives on the project steering group. An initial set of themes, based on the data summaries, was generated by TM in a Microsoft Word document. This was refined through cross-case comparison, in-depth discussion between TM, MM and JM and further testing against the summaries and NVivo data. The model was developed alongside this by, first, summarising the outcomes described by interviewees for inclusion in the model. Then, factors that appeared to contribute to the outcomes were identified and tested against the data, to develop the model. Shapes and arrows were used creatively to model contingencies and dynamic relationships, in accordance with established guidance.[24 25] The aim was not to provide a precise causal mapping of the implementation system but to provide a grounded, summary account of the roles, activities and contributory factors for local-level outcomes across the diverse PCN settings included in the sample. An early draft of the model was presented to the project steering group and was subsequently revised, based on the feedback received.

### Patient and public involvement

Due to the primary focus of the study on senior staff implementation of a national policy, no patients were recruited. The study sits within a research programme that features an experienced Patient and Public Involvement group who were consulted throughout the research process.[30] Patient representatives on the project steering group took part in discussions about findings.

### FINDINGS

Interview data on the early integration of the new clinical pharmacy workforce were summarised in two overarching themes:

1. Local organisational innovations of a national policy under conditions of uncertainty.
2. Local multiprofessional decision-making on clinical pharmacy workforce integration and initial task assignment.

### Theme 1: local organisational innovations of a national policy under conditions of uncertainty

Although a phased implementation of the PCN package was planned (with PCNs forming first, followed by the gradual expansion of the clinical pharmacy workforce), PCNs were still forming at the time of data collection and key decisions pertaining to PCN formation extended into the integration of the clinical pharmacy workforce. Indeed, in order to qualify for funding for the new workforce, GP practices had to first form a PCN. This required the appointment of a CD and decisions about size, membership and operational model, the latter determining how the new clinical pharmacists would be

employed. These decisions involved significant uncertainties, trade-offs and risks and were subject to ongoing debate.

The CD role was widely identified, by both CDs and the SPs in the sample, as being vital to building relationships across PCNs, ironing out emergent challenges (such as finding desk space for the new clinical pharmacy workforce) and making the case to primary care colleagues for participation in the PCN. The CDs within the sample were, however, new to a leadership role and reported many challenges. The publication of a significant amount of national, policy documentation to be read and implemented at speed, alongside other complex national policies, made it difficult for CDs to understand the details of what was expected locally. The process of forming a PCN was particularly challenging where GP practices were new to collaborating across organisational divides. One CD, who described their PCN's setup as being 'fraught with contention', had recently resigned from post, citing workload concerns (CD-1).

Some GPs and practice staff were reported to be sceptical of the workload implications of the PCN policy because the new clinical pharmacists would have to be trained and supported in the role. Some previous, negative experiences of working with pharmacists in primary care heightened such concerns. Local scepticism could make PCN participation a hard 'sell' for CDs, especially as the early impact of the PCN policy had been to increase rather than decrease workloads as had been intended:

> I don't see any reduction in anyone's workload at the moment. If anything, we're getting busier. Its: 'We've got lots of staff but we're not sure what they're doing'. That's what they're telling me. Obviously, it's me that needs to sell that to them but it's very hard to try and convince them. (CD-2)

A further stumbling block was uncertainty about the extent that PCNs could decide on priorities for the new clinical pharmacy workforce. Some planned PCN clinical pharmacy services, notably the Structured Medication Review, were entirely new to primary care. The perception locally was that they would therefore not reduce current practice workloads, leaving little incentive to support the PCN policy:

> If it's 'Would you like to use a new workforce to do additional work as well as what you're already doing?', there's no incentive there (SP-1b)

NHS England's decision to permit a period of prolonged flexibility regarding the utilisation of the new clinical pharmacist workforce, due to the COVID-19 pandemic,[9 31] was widely welcomed. However, there was widespread concern this would give way in favour of a performance management approach in future:

> Unfortunately, there's a bit of uncertainty about what NHS England are going to want for their money…

[attached] metrics don't necessarily align with practice priorities (SP-1)

Some PCNs decided on operational models that implied least risk for member practices, despite the presence of trade-offs, because of the unclear benefits of the clinical pharmacy workforce. Operational models for employing this new workforce ranged from direct employment by individual GP practices or by a single, typically large general practice holding the employment contracts for all general practices within a PCN (ie, the 'Lead Practice' model); to indirect employment via a GP Federation (ie, a pre-existing, formal grouping of GP practices) or contracted from a private provider. While direct employment was widely seen to enhance the capacity of PCNs to manage and direct the new clinical pharmacy workforce, the risk of employment liabilities increased:

> There's a lot of anxiety about liabilities for practices because GP partnerships are not limited liability companies…it could be a risk if you end up in an employment tribunal or something like that with risk to the partners…There was a lot of uncertainty about the structure of the networks, how legally it would work, how a contractor would work with each practice and so on, that practices felt a bit uneasy….To be truthful, we're still negotiating how that's going to work. (CD-1)

PCNs that mapped onto pre-existing GP Federations could initiate processes for recruiting the new clinical pharmacists quickly. This meant they could recruit pharmacists that they considered most suited to the role, in what was widely reported to be a tight labour market. One CD of a GP Federation-aligned PCN reported, unusually, having to 'bat them away' (CD-3a). However, the GP Federation model received contrasting appraisals in the sample, which probably reflects local variation in the capability of these organisational forms to serve as policy vehicles.[32] Appraisals included concerns about a lack of operational autonomy and the quality of GP Federation support. One SP argued that employment via GP Federation created an 'identity problem' for new clinical pharmacists once in post:

> Unfortunately, our GP Federation isn't a strong one. They don't give any support other than payroll, and they take a management fee…It leaves people with an identity problem. Their employer is the GP Fed but it doesn't do anything other than pay their salary. (SP-1b)

The absence of an optimal operational model meant that PCN decisions about employment arrangements were subject to ongoing debate and experimentation that complicated the early utilisation of the new clinical pharmacists once recruited. While some GPs and practice managers were reported to be sceptical initially, however, positive experiences of the clinical pharmacists, once in post, could increase support for the PCN policy, with

subsequent implications for local choices about operational models. One CD reported that local practices were seeking to pay for additional PCN clinical pharmacists because successful, early integration had given practices more 'confidence':

> It's given practices more confidence to almost try before they buy into it, and take on the employment risks and rights themselves (CD-3b)

In some PCNs, the new clinical pharmacists' contributions to the COVID-19 response could shift local opinion. One CD observed an increase in support for direct employment due to their clinical pharmacist's crucial role in the primary-care-led vaccine programme, which was rolled out during the study period. The programme had, unexpectedly, presented formative opportunities for their PCN pharmacy team which contrasted with a neighbouring PCN that had decided against direct employment:

> It became a catalyst, actually, to integrate the team into the PCN. There's nothing else except vaccine: you eat, sleep, dream about vaccine. It would have been impossible without having a pharmacist in the PCN. It made such a difference compared to our neighbouring PCN who hasn't had their own inhouse pharmacist. (CD-2b)

### Theme 2: local multiprofessional decision-making on clinical pharmacy workforce integration and early task assignment

A long-term aim of the PCN policy was to implement, at scale, a patient-facing clinical pharmacy role, although there was recognition that the new clinical pharmacy workforce would take time to develop the requisite skills for this. In order to qualify for ARRS funding, newly recruited clinical pharmacists had to undertake formal training for the role, delivered by the Centre for Pharmacy Postgraduate Education. This 18-month pathway provided them with an introduction to primary care and training in patient-facing, consultation skills. There was also an expectation that each clinical pharmacist would be assigned a GP supervisor to develop and support them in the role. However, this took place within a context of staff and service pressures and the COVID-19 pandemic. While the primary care-led vaccine programme presented formative opportunities for some PCN pharmacy teams, constraints on in-person interaction were widely held to slow the development of a patient-facing role. One SP, who had had 18 years of experience of primary care pharmacy, warned that policymakers' unrealistic expectations were setting up clinical pharmacists to fail:

> From my understanding of advanced practice, I wouldn't put myself [yet] where they're selling the clinical pharmacists will be in 18-months' time. You're setting them up to fail (SPX-9)

The SPs in the sample had roles planning PCN pharmacy services and assisting with recruitment, management and supervision. They highlighted the unique experience, knowledge and skills they brought to this role, including their understanding of primary care pharmacy and, indeed, medications, which GPs do not necessarily have. The supervision they provided was considered complementary to that of GPs as it focused on medication-related questions, with GPs answering diagnostic-related questions. Having gone through a similar process to the new recruits, moreover, they considered themselves well-placed to build confidence and adapt early task assignment to each individual:

> We decide, almost through shared decision-making with them, what level they're at, sign off their competencies, build it up slowly, give them, not boring tasks but simpler things…And then, from building your confidence, we'll develop you into the clinical pharmacist that…we want in our team (SPX-1)

SPs' autonomy to shape the integration of the clinical pharmacists differed markedly. The extent that CDs were 'pro-pharmacy' or knowledgeable of the clinical pharmacist role was reported to vary:

> If I compare our PCN against some of the other[s]… within our locality, the Clinical Directors have been pro-pharmacy…The other Clinical Directors have struggled to grasp what a clinical pharmacist can do (SPX-8)

Many of the SPs shared concerns about 'GP led' PCNs (SPY-3b) which lacked sufficient senior pharmacy input and were purportedly employing clinical pharmacists solely as a means of reducing practice workloads rather than balancing practice priorities with strategic PCN objectives. Although many expressed sympathies with GPs about longstanding workload pressures in primary care, the SPs were concerned that, in these PCNs, the initial tasks being assigned were not aligned with the skills, experiences or aspirations of the new clinical pharmacists. Examples of early skill over-utilisation and under-utilisation were reported, with possible implications for staff well-being, retention and patient safety. For example, one SP described how early, GP-led task assignment may have contributed to medication-related errors:

> We've just had a massive issue switching everyone from Warfarin to NOACs [novel oral anticoagulants]. The pharmacists didn't really have any training in it, so we've now got people on NOACs that shouldn't be on NOACs. I've found two that need to back on Warfarin. So, they need to have training before they do anything and that seems to be a mismatch (SP-2)

A more common concern was that new clinical pharmacists were being assigned basic administrative tasks, described as 'all the, kind of, mop up jobs for the GP' (SPX-1). The prospect of path dependency in role development was also widely recognised, with some PCNs described as being 'stuck in a rut' (SPY-2).

PCN leaderships developed contrasting approaches for integrating clinical pharmacists into general practices. Some PCNs opted to have clinical pharmacists float around individual general practices. Under this model, PCN priorities predominated, with the PCN pharmacy team offering set services to practices such that the problem of skill under-utilisation or over-utilisation in early task assignment was avoided. A reported weakness of this model was that the new clinical pharmacists could struggle to form working relationships with GPs and other general practice staff. A different approach was to assign a clinical pharmacist to each individual practice and attempt to balance PCN and practice requirements. This model could result in practice priorities predominating in early task assignment but a reported benefit was that the clinical pharmacists could integrate better into individual general practices and form stronger relationships with general practice staff. As with PCN choice of operational model, approaches for integrating the clinical pharmacists into general practices were subject to change as learning about the roles was acquired locally, and increasing numbers of clinical pharmacists were employed. One SP, critical of their PCN leadership's approach, saw an opportunity to rethink this as their CD stepped down:

I think now is the opportunity, with this new Clinical Director coming in, to pull them out, do more training with them and change and direct the role, because they're not practice pharmacists, they're not care home pharmacists, they're not CCG pharmacists: they're PCN pharmacists (SP-2)

### The PCN policy emerging in practice

The findings encapsulate key issues relevant to the early integration of the clinical pharmacist workforce within newly forming PCNs; figure 1 provides, in diagrammatical form, a cross-theme summary model of policy emergence and adaptation on the ground, based on interviewee accounts.

To the left of the model are those aspects of the national PCN policy that were observed to be stimulating local action and decision-making on the ground, notably PCN funding and the DES contract. The two overlapping circles in the middle represent the core, intertwined processes involved in local implementation: PCN formation and clinical pharmacist integration. The initial plan was for PCNs to form in the first year of the policy and for PCN pharmacy teams to expand thereafter: the overlap, along with the double-ended, curved arrows linking the circles, indicates that PCNs were not developing in such a linear process. Rather, PCNs were still being formed more than 2 years into policy implementation and crucial decisions about operational models extended long into the process of integrating the clinical pharmacists into PCNs.

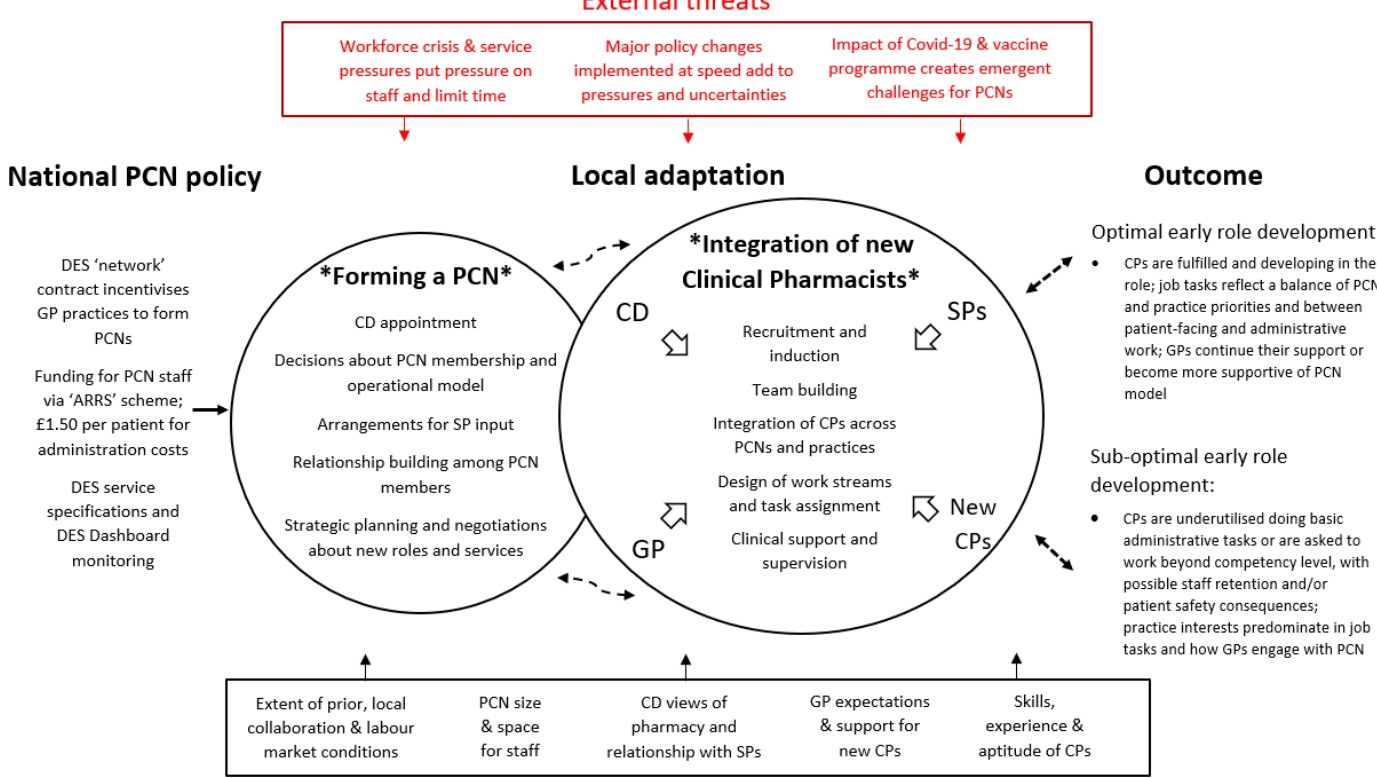

**Figure 1** Early Implementation of the PCN clinical pharmacy role in newly formed PCNs. ARRS, Additional Roles Reimbursement Scheme; CD, Clinical Director; DES, Directed Enhanced Services; GP, General Practitioner; PCNs, Primary Care Networks; SP, Senior pharmacist; CP, Clinical Pharmacist.

This second process was complicated by the fact that some GPs and practice managers were not sure whether to support it and the absence of an optimal operation model meant that, in some PCNs, clinical pharmacists were being integrated into organisations in a state of flux. Local learning about the possible contribution of clinical pharmacists could, however, lead to decisions pertaining to PCN operational models being revisited, with a view to more optimal integration in future.

The core outcomes, at this stage of the PCN policy, are summarised to the right of model: outcomes were categorised as optimal or suboptimal, based on interviewees' evaluations of clinical pharmacist integration. The double-ended arrows between the outcomes and the circle representing clinical pharmacist integration raise the possibility of feedback loops where an initial success paves the way for further successes as PCN members see the potential of the new workforce. The converse is also illustrated, as early failures are constraining and invite path dependency in role development. Below, the 'moderating factors' are candidate explanatory factors, based on this analysis, for variations in the outcomes across the PCNs in the sample. The external threats identified at the top of the model comprise factors reported to impede progress across PCNs generally, although more advanced PCN pharmacy teams could mitigate their effects or, as in the case of the primary care-led vaccine programme, create opportunities out of them.

## DISCUSSION

This paper contributes to scholarship on NHS primary care restructuring and the 'mid-level' professional roles that are emerging globally.[8 9] In particular, it highlights the challenges involved in scaling up new roles across a health system, here a primary care, clinical pharmacy role, being implemented via PCNs, as part of a major policy effort to embed elements of international best practice in the English NHS. The contribution includes examples provided, by SPs, of clinical pharmacists being over-utilised and under-utilised in their early task assignment, with the allocation of excessive basic administrative tasks being a widespread concern. The complex systems modelling exercise revealed potential for path dependency in role development, implying a significant challenge to long-term PCN objectives. This is consistent with a recent report from The King's Fund, which highlights considerable variation in emerging PCN practice, amidst a lack of shared, local understanding about the purpose of the new PCN roles.[20] This emerging picture of undesirable variation in PCN policy implementation suggests the cogency of concerns voiced earlier on in the policy cycle that policymakers were being overambitious about what could be achieved in a challenging environment.[17 33 34]

Indeed, Checkland et al's interview study of national policymakers and stakeholders in the PCN policy in its formative stage revealed contrasting policy objectives that, the authors warned, were likely to complicate implementation.[17] Notable among these was a tension between the objective of supporting primary care via investment and the objective of facilitating the design and delivery of new services across organisational boundaries. Checkland et al[17] concluded that stronger temporal sequencing of implementation, focusing on the development of primary care first before the delivery of new services, would maximise the likelihood of success. Our findings attest to the clear sightedness of Checkland et al's analysis. We found that local decisions about PCN formation and, in particular, how to employ the clinical pharmacists, were complex and characterised by uncertainties, trade-offs and risk. They extended into the integration of the clinical pharmacists, with potential implications for long-term role development. Somewhat ironically, COVID-19 enforced a stronger version of temporal sequencing, similar to that recommended by Checkland et al,[17] as the delivery of PCN services was paused such that a new workforce was available to assist in the primary care-led vaccine programme.[9 31] PCNs appear to have had varying degrees of success in integrating the clinical pharmacist workforce under these conditions however. Immediate research and policy implications of the analysis are discussed next, followed by implications for wider debates about NHS governance:

*Research implications*: the unfolding course of events in restructuring primary care and optimising the contribution of the expanded clinical pharmacist workforce deserve to be studied closely, as do other national policy innovations largely or entirely contingent on local priorities for delivery. This study's focus on the clinical pharmacist workforce complements emerging research into PCN healthcare role development.[20] The key research implications here concern the workforce development needs of clinical pharmacists, how these are met in practice and with what impacts on the roles that emerge.

*Policy implications*: the picture painted above of undesirable variation in PCN policy implementation suggests that a comprehensive package of support, as outlined in The King's Fund report,[20] is required, if PCNs are to expand the primary care workforce effectively and serve a prominent, coordinating role in the emerging NHS architecture. Support from the new NHS partnership structures known as Integrated Care Systems, also still forming, will be important, including organisational and leadership skills development. An extension of the ARRS funding, beyond the 5 years planned, would provide struggling PCNs with more time, and may alleviate some of the concern about employment liabilities. This study has also highlighted a particular need for a clearer policy and professional framework for senior PCN pharmacists: their input, though variable across PCNs, was shown to be vital to the successful integration of clinical pharmacists and this is likely to continue through role development.

*Governance implications*: the challenges identified in the study pertaining to PCN formation resonate with the interim findings of a National Institute for Health Research funded study of PCNs which found that their contractual nature presented a major implementation barrier.[16] Here, the apparent absence of an optimal PCN operational model and associated local concern about employment liabilities was shown to complicate the early integration of the clinical pharmacists. The extent of setup effort required implies that the PCN model is currently failing to realise savings in transaction costs, a key source of efficiency in public administered (as opposed to private or market-based) health systems.[35] An alternative, public sector administration approach to primary care development would have been to integrate GP practices within the NHS and to expand the primary care workforce via direct employment by statutory bodies.[36] Such an approach may have reduced local exposure to employment liabilities and streamlined the ask of GPs and practice staff to focus on clinical pharmacist integration and role development.

**Contributors** JM, MM and DS designed the study. TM and MM conducted the interviews and analysed the data. All authors made substantial contributions to the theorisation of the data through group discussions and paper development. TM conceptualised and led the write up of the paper. All authors contributed to refining the themes and the model. DS contributed policy expertise and BG contributed methodological expertise. JM is responsible for the overall content as the guarantor.

**Funding** This research was funded by the National Institute for Health Research [NIHR] PGfAR [RP-PG-0216-20002].

**Disclaimer** The views expressed are those of the authors and not necessarily those of the NIHR or the Department of Health and Social Care.

**Competing interests** None declared.

**Patient and public involvement** Patients and/or the public were involved in the design, or conduct, or reporting or dissemination plans of this research. Refer to the Methods section for further details.

**Patient consent for publication** Not applicable.

**Ethics approval** This study was approved by NHS Health Research Authority/REC approval was obtained for the study. The REC reference is as follows: 20/HRA/1482.

**Provenance and peer review** Not commissioned; externally peer reviewed.

**Data availability statement** No data are available. No data are available. This study has not received ethical approval to share confidential data with any third party other than the study research team.

**ORCID iDs**
Thomas Mills http://orcid.org/0000-0003-2599-8930
Mary Madden http://orcid.org/0000-0001-5749-2665

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
