## [Reviewer comments · BMJ Open]

ARTICLE DETAILS

TITLE (PROVISIONAL)	The integration of a clinical pharmacist workforce into newly forming Primary Care Networks: a qualitatively-driven, complex systems analysis
AUTHORS	Mills, Thomas; Madden, Mary; Stewart, Duncan; Gough, Brendan; McCambridge, Jim

VERSION 1 – REVIEW

REVIEWER	Patel, Nilesh University of Reading, Pharmacy
REVIEW RETURNED	08-Jul-2022

GENERAL COMMENTS	Thank you for providing a timely and very interesting article on the clinical pharmacy workforce integration within PCN's. This is an important area to research and inform policy makers and PCN workforce about. There are some areas that need clarification. These are fairly minor but I hope will add further clarity to the article. Methods section - I feel this needs a bit more detail about the qualitative approach and research paradigm- Can more information be provided on how participants were identified (you mention opportunistic sampling and snowballing but how to you know who to contact and did what social media did you use?), what information was sent and at what point did you gain consent?- Page 7, line 6. Did TM and MM carry out the interviews together or separately?- Can you please say some more about complex systems modelling. For someone who is not familiar with this, it would be important to know how it is used and justify why you are using it.- NVIVO vs NVivo- Page 7, line 22. Spell out PPI Overall the paper is well written and i have no comments regarding the results and discussion.
--

REVIEWER	Wilson, Tim Oxford Centre for Triple Value Healthcare
REVIEW RETURNED	12-Jul-2022

GENERAL COMMENTS	Page 4 Future studies would benefit from interviewing other general practice staff (e.g. receptionists, practice managers and practice nurses) There is a risk of group think as it is highly likely that interviewees had been sharing views prior to the study.
--

	Page5 Line 11- Gatekeeper is an overused and rather simplistic term leading to a misunderstanding of what a GP does. Starfield uses a better term- coordinator of care. The authors are perhaps too young to recall that pharmacists were employed by GP fundholders for medicines optimisation. Page 6 Sections b) and c)- the authors omit to mention where these interviewees were working. Lines 42 and 48- the editors will need to decide whether the very opportunistic and pragmatic approach to sampling is a barrier to publication. I do not believe that the data gleaned would have been different had the sampling technique been more structured. Personally, I did not find the graphic on page 19 helpful. It could be simplified.
--	--

REVIEWER	Bailey, Simon University of Kent, Centre for Health Services Studies
REVIEW RETURNED	18-Jul-2022

GENERAL COMMENTS	BMJ Open review 18/7/22 This is a very clearly written and warranted paper that looks at integrating clinical pharmacists into newly forming primary care networks, which is an important issue around the objectives and implementation of the policy as well as the longer term aim of diversifying the primary care workforce. For the most part the paper is well presented and well argued, however, it is let down by discussion, which needs to demonstrate how it develops the findings presented as well as existing literature. Literature review is a bit brief in places and some old literature used (e.g. there is a 2018 Laurant et al. Cochrane review which supercedes the 2005 one). There is also literature on early implementation of PCNs – some of this is included in discussion (Baird, Pettigrew) but should be summarised at the start, and there are some gaps: Checkland, K., Hammond, J., Morciano, M., Warwick-Giles, L., Lau, Y.S., Bailey, S. and Sutton, M., 2021. Primary Care Networks: exploring primary care commissioning, contracting, and provision. NIHR Policy Research Unit in Health and Social Care Systems and Commissioning. Warwick-Giles, L., Hammond, J., Bailey, S. and Checkland, K., 2021. Exploring commissioners' understandings of early primary care network development: qualitative interview study. British Journal of General Practice, 71(710), pp.e711-e718. Parkinson, S., Smith, J. and Sidhu, M., 2021. Early development of primary care networks in the NHS in England: a qualitative mixed-methods evaluation. BMJ open, 11(12), p.e055199. Methods. In the datasets b) and c) it isn't clear where these PCNs were or if there was overlap with those involved as part of a). Reading on this detail is provided below but it would be clearer to include it within the description of each. Findings Discussion of GP federations should note the wide variation in how formal or operational federations were prior to PCN policy being implemented – i.e. these are not legitimate policy vehicles in many areas. There is literature on this: McDonald, R., Riste, L., Bailey, S., Bradley, F., Hammond, J., Spooner, S., Elvey, R. and Checkland, K., 2020. The impacts of GP federations in England on practices and on health and social care
--

	interfaces: four case studies. Health Services and Delivery Research, 8:11 https://doi.org/10.3310/hsdr08110 I really like the attempt to visualise the findings related to the emerging policy – this conveys the complexity well. Discussion The discussion would benefit from being more focused, it is a bit woolly at present, it is not clear how the findings are being used to direct the discussion and implications both of which would also benefit from more systematic structuring around existing literature. For example, the governance implications are sensible but it's not clear how you've got from your findings to this. The last two sentences in the governance implications are both rather speculative and again bear no obvious relation to what has come before.
--	---

VERSION 1 – AUTHOR RESPONSE

Reviewer: 1

Dr. Nilesh Patel, University of Reading

Comments to the Author:

Thank you for providing a timely and very interesting article on the clinical pharmacy workforce integration within PCN's. This is an important area to research and inform policy makers and PCN workforce about. There are some areas that need clarification. These are fairly minor but I hope will add further clarity to the article.

Methods section

- I feel this needs a bit more detail about the qualitative approach and research paradigm – **Thanks – we have responded to your specific points and feel this has strengthened our presentation of the approach we took**

- Can more information be provided on how participants were identified (you mention opportunistic sampling and snowballing but how to you know who to contact and did what social media did you use?), what information was sent and at what point did you gain consent? – **We now describe this in greater detail**

- Page 7, line 6. Did TM and MM carry out the interviews together or separately? **Separately – we now note this.**

- Can you please say some more about complex systems modelling. For someone who is not familiar with this, it would be important to know how it is used and justify why you are using it. **Thanks – we now provide detail of the complex systems approach when outlining the research question at the end of the introduction.**

- NVIVO vs NVivo – **Changed to NVivo**

- Page 7, line 22. Spell out PPI – **Changed**

Overall the paper is well written and i have no comments regarding the results and discussion. – **Thanks**

Reviewer: 2

Dr. Tim Wilson, Oxford Centre for Triple Value Healthcare

*** This reviewer has included an attachment alongside their review. Please find it attached to this email *** **This has been incorporated into the findings section**

Comments to the Author:

Page 4

Future studies would benefit from interviewing other general practice staff (e.g. receptionists, practice managers and practice nurses)

There is a risk of group think as it is highly likely that interviewees had been sharing views prior to the

study. Thanks – we now mention, in the strengths and limitations section, that broadening out to include other general practice staff would allow exploration of diverse perspectives and multi-disciplinary working.

Page 5

Line 11- Gatekeeper is an overused and rather simplistic term leading to a misunderstanding of what a GP does. Starfield uses a better term- coordinator of care. We now refer to GP's role of coordinating patient care.

The authors are perhaps too young to recall that pharmacists were employed by GP fundholders for medicines optimisation. We now include mention of the GP Fundholding scheme.

Page 6

Sections b) and c)- the authors omit to mention where these interviewees were working. We now include a separate sentence describing where the interviewees were recruited.

Lines 42 and 48- the editors will need to decide whether the very opportunistic and pragmatic approach to sampling is a barrier to publication. I do not believe that the data gleaned would have been different had the sampling technique been more structured.

Personally, I did not find the graphic on page 19 helpful. It could be simplified. The development of the model was an integral part of the analytic process and involved input from senior stakeholders and PPI representatives. We do not think it can be simplified without obscuring the complexity of PCN policy implementation. Furthermore, we have included detailed, written findings in the themes so that readers can read them if they do not find the model helpful. We also note Reviewer 3's comment that it "conveys the complexity well".

Reviewer: 3

Dr. Simon Bailey, University of Kent

Comments to the Author:

BMJ Open review 18/7/22

This is a very clearly written and warranted paper that looks at integrating clinical pharmacists into newly forming primary care networks, which is an important issue around the objectives and implementation of the policy as well as the longer term aim of diversifying the primary care workforce. For the most part the paper is well presented and well argued, however, it is let down by discussion, which needs to demonstrate how it develops the findings presented as well as existing literature. Literature review is a bit brief in places and some old literature used (e.g. there is a 2018 Laurant et al. Cochrane review which supercedes the 2005 one). There is also literature on early implementation of PCNs – some of this is included in discussion (Baird, Pettigrew) but should be summarised at the start, and there are some gaps:

Checkland, K., Hammond, J., Morciano, M., Warwick-Giles, L., Lau, Y.S., Bailey, S. and Sutton, M., 2021. Primary Care Networks: exploring primary care commissioning, contracting, and provision. NIHR Policy Research Unit in Health and Social Care Systems and Commissioning.

Warwick-Giles, L., Hammond, J., Bailey, S. and Checkland, K., 2021. Exploring commissioners' understandings of early primary care network development: qualitative interview study. British Journal of General Practice, 71(710), pp.e711-e718.

Parkinson, S., Smith, J. and Sidhu, M., 2021. Early development of primary care networks in the NHS in England: a qualitative mixed-methods evaluation. BMJ open, 11(12), p.e055199.

Thanks – this has been extremely helpful to the literature review aspect of the paper. We have incorporated a new paragraph in the introduction that summarises the articles you have suggested. Methods.

In the datasets b) and c) it isn't clear where these PCNs were or if there was overlap with those involved as part of a). Reading on this detail is provided below but it would be clearer to include it within the description of each. We have made it clear that sub-studies a and b included diverse PCNs from across England and that there was some overlap. We state that, in total, 28 senior PCN staff were interviewed across 17 PCNs.

Findings

Discussion of GP federations should note the wide variation in how formal or operational federations were prior to PCN policy being implemented – i.e. these are not legitimate policy vehicles in many areas. There is literature on this:

McDonald, R., Riste, L., Bailey, S., Bradley, F., Hammond, J., Spooner, S., Elvey, R. and Checkland,

K., 2020. The impacts of GP federations in England on practices and on health and social care interfaces: four case studies. Health Services and Delivery Research, 8:11 <https://doi.org/10.3310/hsdr08110>. Thanks - this is an important point and we have included it (and the reference) in the findings section.

I really like the attempt to visualise the findings related to the emerging policy – this conveys the complexity well. Thank you

Discussion

The discussion would benefit from being more focused, it is a bit woolly at present, it is not clear how the findings are being used to direct the discussion and implications both of which would also benefit from more systematic structuring around existing literature. For example, the governance implications are sensible but it's not clear how you've got from your findings to this. The last two sentences in the governance implications are both rather speculative and again bear no obvious relation to what has come before. Thanks – We have grounded the discussion more fully in the findings and existing literature throughout. For example, the governance implications paragraph has been amended to focus more centrally on the issue of the contractual nature of PCNs and the employment of the clinical pharmacists. To avoid this appearing speculative, the final sentence highlights implications for clinical pharmacist integration and role development.

VERSION 2 – REVIEW

REVIEWER	Bailey, Simon University of Kent, Centre for Health Services Studies
REVIEW RETURNED	28-Sep-2022
GENERAL COMMENTS	Thank you for your attention to revisions. I believe this is a useful addition to the literature and have no further suggestions for changes.